# CELL2TEXT: MULTIMODAL LLM FOR GENERATING SINGLE-CELL DESCRIPTIONS FROM RNA-SEQ DATA

## ABSTRACT

Single-cell RNA sequencing has transformed biology by enabling the measurement of gene expression at cellular resolution, providing information for cell types, states, and disease contexts. Recently, single-cell foundation models have emerged as powerful tools for learning transferable representations directly from expression profiles, improving performance on classification and clustering tasks. However, these models are limited to discrete prediction heads, which collapse cellular complexity into predefined labels that fail to capture the richer, contextual explanations biologists need. We introduce Cell2Text, a multimodal generative framework that translates scRNA-seq profiles into structured natural language descriptions. By integrating gene-level embeddings from single-cell foundation models with pretrained large language models, Cell2Text generates coherent summaries that capture cellular identity, tissue origin, disease associations, and pathway activity, generalizing to unseen cells. Empirically, Cell2Text outperforms baselines on classification accuracy, demonstrates strong ontological consistency using PageRank-based similarity metrics, and achieves high semantic fidelity in text generation. These results demonstrate that coupling expression data with natural language offers both stronger predictive performance and inherently interpretable outputs, pointing to a scalable path for label-efficient characterization of unseen cells.

## 1 INTRODUCTION

Single-cell RNA sequencing (scRNA-seq) enables measurement of gene expression at the resolution of individual cells, opening new possibilities for mapping tissue organization, reconstructing developmental processes, and studying disease at a fine-grained scale. Despite this potential, interpretation of scRNA-seq data still depends heavily on annotation, where cells are labeled by type, state, or function using marker genes and expert knowledge. This process is both slow and subjective, and it does not scale to the millions of cells now routinely generated by modern experiments. As datasets continue to grow, there is a pressing need for computational frameworks that go beyond predefined categories and provide richer, biologically grounded descriptions of cellular identity and function.

A range of computational methods have been developed to automate annotation, with recent attention focused on large foundation models. Approaches such as Geneformer (Theodoris et al., 2023b) and scGPT (Cui et al., 2024a) generate powerful embeddings of gene expression profiles and have shown promise across several tasks. However, these methods typically require classification heads to map embeddings onto predefined categories. This reliance introduces important limitations: new cell types, states, or biological contexts demand additional training or fine-tuning, a process that is often computationally intensive and impractical for many biology labs. As a result, even though embeddings capture rich structure in the data, their utility remains constrained by static label spaces and specialized model development workflows. Together,

these issues point to the need for a more flexible approach that can generalize beyond static labels and deliver richer, more contextual descriptions of cellular identity and function.

In this work, we introduce Cell2Text, a multimodal generative framework that converts scRNA-seq profiles into natural-language descriptions. Our contributions can be summarized as follows:

- We introduce **Cell2Text**, a **multimodal generative framework** that aligns gene-level embeddings with instruction-tuned language models, producing interpretable, context-rich natural language descriptions.

- We demonstrate that Cell2Text achieves **competitive performance** across cell type, tissue, disease, and pathway classification tasks, with ontology-aware evaluation revealing that even incorrect predictions maintain high biological relevance.

- We show that Cell2Text generates **high-quality natural language descriptions** with exceptional semantic similarity and biological soundness, demonstrating the model's ability to produce scientifically meaningful and interpretable cellular characterizations.

- We construct a **large-scale multimodal dataset** of 1M cells from CELLxGENE, enriched with ontology terms, tissue and disease metadata, and pathway annotations. This dataset supports cross-modal training and evaluation at a scale not available in existing resources.

## 2 RELATED WORK

Single cell analysis is rooted from traditional experimental biology, where cell types can be distinguished via lab assays and known markers. The raise of single-cell RNA sequencing (scRNA-seq) provided a way to numerically profile cells, but the high-dimensional data remains the obstacle. Early scRNA-seq studies relied on unsupervised clustering with manual annotation of clusters based on biomarkers (Xie et al., 2021; Ianevski et al., 2022; Cheng et al., 2023), which is labor-intensive and requires expert knowledge (Ranjan et al., 2021; Kim et al., 2023). Large consortium efforts including the Human Cell Atlas (Human Cell Atlas Consortium, 2017; Hon et al., 2018; Abhulimen, 2024) produced reference, yet mapping new cells to these references demands careful handling of noise and batch effects (Kang et al., 2021; Lotfollahi et al., 2022; Luecken et al., 2022). While current scRNA-seq pipelines are mostly used for identifying cell types and lineage trajectories (Lotfollahi et al., 2024; Li et al., 2025), the interaction of the cell with the environment remains limited.

To automate cell annotation, many computational methods formulate it as a reference-based classification. SCMAP (Kiselev et al., 2018), SingleR (Aran et al., 2019) and many other straightforward approaches (Huang et al., 2021; Pasquini et al., 2021) match cells to reference profiles with cosine similarity, while CHETAH (de Kanter et al., 2019) evaluates how well a cell fits the expression distributions of known types. As complex non-linear patterns are ignored with barely pairwise similarity (Chang et al., 2024), other methods explicitly train machine learning classifiers on annotated datasets. scPred (Alquicira-Hernandez et al., 2019) uses a support vector machine, SingleCellNet (Tan & Cahan, 2019) applies an ensemble of decision trees, and the Seurat toolkit (Stuart et al., 2019) implements the label transfer using reference atlas integration. Other works (Wang et al., 2021b; Lewinsohn et al., 2023; Bhadani et al., 2023) construct graphs of cells then propagate label information via diffusion, improving efficiency on large scale data. Common limitation of these approaches lies in the lack of curated marker gene lists (Pullin & McCarthy, 2024), improper handling of batch differences (Luecken & Theis, 2019; Zappia et al., 2025), and inability to leverage higher-order gene-gene interactions (Wang et al., 2021a).

Another direction leverages foundation models as powerful feature extractors for classification. These transformer encoders are much larger and are pre-trained on massive single-cell datasets. scBERT (Yang et al., 2022) adopts the BERT text encoder structure, and Geneformer(Theodoris et al., 2023a) pre-trained on

around 30 million human single-cell transcriptomes, both demonstrating good performance on diverse pre-diction tasks with a pooling layer. scGPT (Cui et al., 2024b) introduces a similar decoder-only structure, while CellWhisperer (Schaefer et al., 2024) applies the CLIP-style contrastive learning to align gene expressions with transcriptions, but used bulk RNA-seq expressions instead of single-cell data. Despite these advances, existing approaches inherently produce only categorical labels or limited annotations. These models also face challenges in scaling, as attaching a classification head requires enumerating all possible categories. Also, the unavoidable pooling layer leads to serious loss of information, as many downstream tasks may rely on subtle details.

These limitations spurred recent interest in Large Language Models (LLMs) that can generate richer and more descriptive outputs. Cell2Sentence (C2S) (Levine et al., 2024) and Cell2Sentence-Scale (C2S-Scale) (Rizvi et al., 2025) pioneer in this direction by converting gene expression data into a natural language sentence where names of top-100 expressed genes are ordered by their expression level in that cell. However, representing cells as plain text gene sentences offers only shallow signals, since the language decoder is not pretrained on such inputs and cannot fully exploit hidden biological patterns encoded in the data. In addition, truncating sequences to this limit constrains the model's ability to capture subtle gene–gene interactions, particularly in lowly expressed regions.

A better way to overcome these issues is to use a pretrained cell encoder along with a pretrained language decoder to better pick up patterns and relationships between genes. Similar ideas have been tested in other areas of biology, as Prot2text (Abdine et al., 2024) and Prot2Text-V2 (Fei et al., 2025) show that pretrained models for protein sequences can produce meaningful test reflecting underlying biology, while ChatNT (de Almeida et al., 2025) investigates such possibility with DNA sequences. Yet for cells, this approach has barely been explored, leaving open the chance to connect pretrained cell representations with language generation in a more biologically informed way.

## 3 METHODOLOGY

Cell2Text goal is to generate comprehensive and accurate natural language descriptions of single cells. These descriptions synthesize crucial information including cell type, associated disease, tissue origin, donor development stage, and active pathways derived directly from gene expression profiles. Our approach combines a specialized cell encoder with a natural language decoder through an adapter mechanism that projects high-dimensional cellular representations into the language model's semantic space, enabling the generation of detailed, biologically meaningful descriptions.

### 3.1 MODEL ARCHITECTURE

#### 3.1.1 CELL ENCODER

The Cell Encoder module represents the first critical component of Cell2Text, responsible for converting raw single-cell gene expression data into meaningful, context-aware embeddings that our language model can effectively interpret. We chose Geneformer (Theodoris et al., 2023b; Chen et al., 2024), a transformer-based model pre-trained specifically on single-cell genomics data on masked gene prediction task, as our cell encoder due to its proven ability to capture the complex relationships between genes. What makes Geneformer particularly well-suited for our task is its training on large-scale single-cell transcriptomic datasets, which allows it to learn gene-gene interactions and produce contextualized gene representations.

We propose a gene-level embedding strategy that differs from the conventional cell-level pooling approach adopted in most prior studies. Instead of compressing each cell into a single embedding vector, which would lose important biological nuances, we extract individual embeddings for each gene within the cell. For a given cell, Geneformer produces a sequence of $N$ gene embeddings, where $N$ corresponds to the number

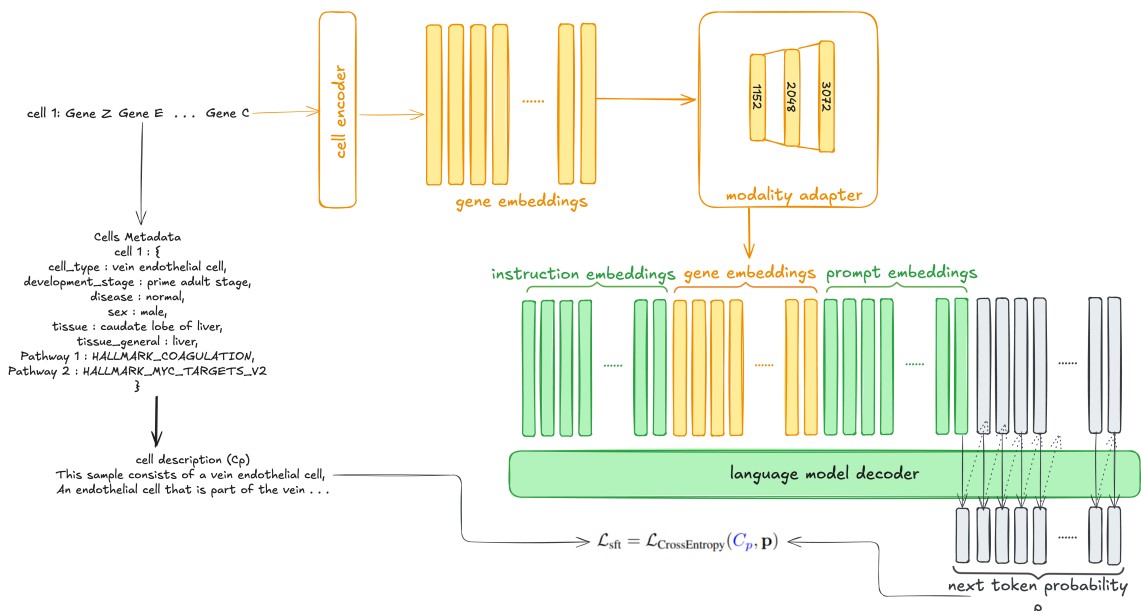

Figure 1: Overview of the Cell2Text framework. The model takes single-cell RNA-seq profiles as input and processes them through a pretrained Geneformer encoder to generate contextualized gene-level embeddings. These embeddings are projected into the semantic space of the language model via a lightweight adapter module, aligning biological signals with linguistic representations. A pretrained, instruction-tuned LLM decoder then generates structured natural language descriptions that capture cellular identity, tissue of origin, disease associations, and pathway activity.

of genes in the sequence. The value of $N$ is fixed based on the maximum context length supported by the encoder, which is 4096 genes for **Geneformer-V2-316M**; this constraint is not limiting since most cells express fewer than 4096 genes, while the highest-expressed genes include most of the biological information. Each of these high-dimensional gene embeddings captures both the expression level and the regulatory context of its corresponding gene within that specific single-cell environment.

This gene-level approach offers several advantages: it preserves the granular transcriptional information that distinguishes different cell types and states, and it provides our downstream language model with a richer, more detailed representation of the single cell.

### 3.1.2 ADAPTER MODULE

We introduce a lightweight adapter module to bridge the dimensional and semantic gaps between the Geneformer's output and the LLM's input embedding space. This module consists of a two-layer feedforward network with non-linear activation that projects each gene embedding to the natural language semantic space. The resulting gene embeddings are L2-normalized to stabilize the training process before being passed to the LLM.

### 3.1.3 NATURAL LANGUAGE DECODER

The Decoder Module constitutes the text generation component of Cell2Text, where contextualized gene embeddings are transformed into cell descriptions using a pretrained LLM. To assess the impact of LLM architecture and scale on description quality, we conducted ablation studies utilizing different publicly available instruction-tuned models:

- **Meta-Llama-3.2-1B-Instruct**: A compact variant of the Meta-Llama series, selected for its efficiency and strong performance on instruction-following tasks, offering a balance between computational demands and descriptive capabilities.

- **Gemma3-4B-it**: Google's 4-billion parameter model featuring a distinctive hybrid attention mechanism with 5:1 interleaving of local sliding window and global self-attention layers, contrasting with Llama's uniform attention architecture. This selection enables evaluation across different architectural paradigms and model scales (4B vs 1B parameters).

### 3.2 TRAINING PROCESS

Given that Geneformer is already pre-trained on extensive single-cell data and its gene embeddings effectively capture biological information and gene-gene interactions, we freeze the Geneformer encoder throughout training to preserve these learned representations. The adapter module remains trainable in both training strategies to enable proper projection from the gene embedding space to the natural language semantic space.

### 3.2.1 FULL FINE-TUNING

We performed full fine-tuning on both Meta-Llama-3.2-1B-Instruct and Gemma3-4B-it models, updating all parameters to adapt the pre-trained LLMs to our domain-specific cell description task. To guide the LLM's generative process and ensure consistent output format, we adopted a specific instruction-following prompt structure that integrates a system message and the contextualized gene embeddings:

> **System**: You are a scientific assistant specialized in cell description predictions. Given the cell sentence embeddings, describe it clearly and concisely in professional language.
> **User**: Sequence embeddings: $H_{g,1}|H_{g,2}|...|H_{g,N}$
> **Assistant**: <CELL DESCRIPTION>

Here, $H_{g,1}|H_{g,2}|...|H_{g,N}$ represents the sequence of gene embeddings from the Cell Encoder, which are projected into the LLM's input embedding space. This structured prompt explicitly defines the task and the desired output format, facilitating the generation of detailed single-cell descriptions.

### 3.2.2 PARAMETER-EFFICIENT FINE-TUNING (PEFT)

Given the substantial computational requirements of full fine-tuning and the specificity of our task, we additionally explored parameter-efficient fine-tuning using Low-Rank Adaptation (LoRA) (Hu et al., 2022) specifically on the Meta-Llama-3.2-1B-Instruct model. LoRA selectively injects trainable low-rank matrices into the transformer architecture's attention mechanism, significantly reducing the number of trainable parameters while maintaining a performance close to full fine-tuning. This approach allowed us to efficiently adapt the pre-trained LLM to our domain-specific task with limited computational resources, while investigating its effect on generation quality compared to the full fine-tuning approach.

## 3.3 CELL TYPE ONTOLOGY PAGERANK SIMILARITY

To evaluate our cell type classification beyond simple accuracy, we employ a similarity metric that captures the biological and hierarchical relatedness between cell types, accommodating "near misses" where a predicted cell type is incorrect but closely related to the true label (e.g., predicting 'T Cell' instead of 'CD4+ T-cell'). Inspired by scCello (Yuan et al., 2024), we utilize the structured knowledge of the Cell Ontology (CL) (Smith et al., 2007) to compute these similarities, enabling a nuanced assessment of how well predictions align with true cell types in terms of biological meaning. Unlike scCello, which applies ontology-based similarity for contrastive learning to train a single-cell encoder, our approach uses this metric to evaluate predictions, providing deeper insight into the model's understanding of cell type relationships and its ability to navigate the hierarchical structure of cell biology.

We model the Cell Ontology as an undirected graph with cell types as nodes and 'is_a' relationships as edges. We use Personalized PageRank (Page et al., 1998) to quantify relatedness. Imagine a random walker starting at a cell type (source node) exploring the graph but biased to return to the starting node (personalization). This bias ensures higher scores for biologically related cell types, like subtypes or parents, while distant ones score lower. For a cell type $c_i$, we compute a Personalized PageRank vector, with personalization centered on $c_i$, where $PPR(c_j|c_i)$ measures relatedness to $c_j$. The similarity score $S(c_i, c_j)$ is:

$$S(c_i, c_j) \propto \log\left(1 + \frac{PPR(c_j|c_i)}{\tau}\right)$$

where $\tau$ scales scores before normalization to $[0, 1]$. The resulting similarity matrix gives identical cell types a score of 1, with scores decreasing with ontological distance. The similarity distribution is heavy-tailed, distinguishing related from unrelated cell types (see Appendix Section C).

## 4 EXPERIMENTS AND RESULTS

To evaluate the performance of Cell2Text, we conducted a series of experiments designed to assess its capabilities in two primary areas: 1) the quality of the natural language descriptions and 2) the accuracy of predicting cellular attributes by parsing generated text. We compare our models against two strong baselines to demonstrate the advantages of our generative approach.

### 4.1 EXPERIMENTAL SETUP

#### 4.1.1 DATASET CONSTRUCTION

We construct a large-scale multimodal dataset that pairs single-cell gene expression profiles with natural language descriptions to enable cross-modal learning between genomic data and biological knowledge. The dataset is derived from the CELLxGENE Census (Program et al., 2024), using a principled sampling strategy (Appendix A.1) to select 1,000,000 cells from 7,331 donors, spanning 783 cell types, 347 tissue types, and 128 disease conditions (Appendix B).

For each cell, we generate structured text descriptions by combining metadata annotations with functional context. Cell type information is enriched using OBO Cell Ontology (Smith et al., 2007) definitions, while biological processes are captured through pathway activity analysis with pySCENIC (Aibar et al., 2017) applied to 34 curated MSigDB Hallmark pathways (Liberzon et al., 2011) (Appendix A.2, Figure 8). For each cell, we identify the two most enriched pathways and translate them into human-readable descriptions of active biological processes using the corresponding MSigDB (Liberzon et al., 2011) definitions. An example of such a description can be found in Appendix A.3.

The resulting descriptions provide interpretable summaries of both cellular identity and functional state, bridging high-dimensional expression profiles with structured biological knowledge. To ensure robust evaluation, we perform donor-level data splitting (80/10/10) to prevent information leakage between training and test sets.

### 4.1.2 BASELINES

For our Cell2Text models, we extract classification labels from the generated descriptions using regular expressions to enable fair comparison with traditional classification approaches. We compare our approach against standard supervised learning methods that directly optimize for classification accuracy. All hyperparameters and training details for the baseline models are provided in Appendix D.

**Disease, Cell Type, and Tissue Classification**. First, we compare against a single linear layer that is trained on top of the corresponding output embedding of the special CLS token from the frozen Geneformer (Geneformer+Head). We also compare against a gradient-boosting model (LightGBM) trained on Geneformer embeddings to perform multi-class classification (LGBM).

**Pathway Classification**. Similarly, we use as a baseline a linear head on top of the frozen Geneformer to output logits for the presence of each of the 34 pathways simultaneously (Geneformer+Head). We also compare with an ensemble model consisting of 34 distinct LGBM classifiers. Each classifier is an expert (classifier of the existence of a specific pathway from the 34 Hallmark pathways), trained to predict the probability of a single pathway's presence based on the cell's embedding. The final prediction is derived by taking the two pathways with the highest probabilities from this set(LGBM).

### 4.2 RESULTS

### 4.2.1 TEXT GENERATION QUALITY

We evaluated the quality of the generated text itself using metrics that assess both lexical overlap and semantic fidelity in Table 1 While exact match rates are low, as expected for generative models that learn to paraphrase, the BLEU and ROUGE scores are high, confirming strong lexical and structural similarity. Most importantly, the semantic scores are outstanding. The exceptionally high BioBERT F1-scores, over 93.9 for fully tuned models, demonstrate that the generated descriptions are not just fluent but are also semantically and scientifically sound within the biomedical domain. A detailed description of the evaluation metrics is provided in Appendix E.1.

Table 1: Evaluation of Cell2Text's text generation capabilities using lexical and semantic metrics. Our models demonstrate strong performance across both dimensions: lexical metrics including Exact Match (Exct), BLEU-2/4 (B-2/B-4), and ROUGE-1/2/L (R-1/R-2/R-L) show strong structural and n-gram overlap with reference texts, while semantic metrics using BERTScore F1 with RoBERTa (RBT-f1) and BioBERT (BBT-f1) reveal remarkable semantic fidelity, with all variants achieving over 93% biomedical semantic accuracy.

| | Exct | B-2 | B-4 | R-1 | R-2 | R-L | RBT-f1 | BBT-f1 |
|---|---|---|---|---|---|---|---|---|
| **Cell2Text-Llama-1B-LoRa** | 5.79 | 77.8 | 73.88 | 82.62 | 76.7 | 79.55 | 95.74 | 93.06 |
| **Cell2Text-Llama-1B** | 7.02 | **80.96** | **77.39** | 84.88 | 79.64 | 81.99 | 96.28 | 93.9 |
| **Cell2Text-Gemma-4B** | 6.73 | 80.91 | 77.38 | **84.99** | **79.79** | **82.17** | **96.32** | **93.93** |

### 4.2.2 Classification from Generated Text

We evaluated the model's ability to accurately predict core cellular metadata: cell type, tissue of origin, and associated disease. As shown in Table 2, all Cell2Text variants consistently outperform both the specialized Geneformer+Head and LGBM baselines.

Despite the strong performance of the Geneformer+Head model, which is explicitly optimized for these tasks, our generative models demonstrate a superior ability to capture and articulate cellular identity. The **Cell2Text-Gemma-4B** model achieves the highest performance in cell type and tissue classification, reaching an accuracy of 77.83% and 73.04%, respectively. This represents a significant improvement of over 10% in cell type accuracy compared to the Geneformer+Head baseline. This result strongly suggests that by training the model to generate coherent descriptions, it learns a far richer and more accurate representation of the cell than what is captured by traditional classification heads.

Table 2: Classification performance of Cell2Text models extracted from generated descriptions compared to other baselines. Our generative approach consistently outperforms all specialized classification methods across cell type, tissue, and disease prediction tasks, demonstrating that learning to generate coherent descriptions leads to superior cellular understanding. Results shown using accuracy and weighted F1-score metrics.

| | cell type | | tissue | | disease | |
|---|---|---|---|---|---|---|
| | accuracy | f1-score | accuracy | f1-score | accuracy | f1-score |
| **Geneformer+Head** | 67.26 | 63.98 | 68.52 | 66.64 | 74.09 | 71.44 |
| **Geneformer+LGBM** | 50.7 | 52.56 | 44.96 | 43.77 | 61.30 | 52.66 |
| **Cell2Text-Llama-1B-LoRa** | 70.90 | 69.28 | 67.97 | 68.03 | 72.71 | 73.44 |
| **Cell2Text-Llama-1B** | 76.91 | 75.88 | **73.35** | **74.02** | **77.84** | **78.46** |
| **Cell2Text-Gemma-4B** | **77.83** | **77.39** | 73.04 | 73.94 | 77.34 | 77.82 |

### 4.2.3 Evaluation with PageRank Similarity

Standard accuracy metrics can be misleading for cell type prediction, as they penalize predictions that are biologically close (e.g., 'CD4+ T-cell' vs. 'T-cell') as harshly as those that are completely unrelated. To address this, we use a PageRank-based similarity score that measures the ontological distance between the predicted and true cell types. We present the results in Table 3. Our fully-tuned Cell2Text models achieve the highest overall similarity scores (85.62% for Gemma-4B), confirming their superior accuracy. This indicates that when these models miss, they are likely to predict a parent or sibling cell type from the ontology, demonstrating a grasp of cellular relationships that simpler classifiers lack.

Table 3: PageRank Similarity (PS) evaluation for cell type classification showing Cell2Text models achieve superior biological understanding.

| Model | Average PS (%) |
|---|---|
| **Geneformer+Head** | 80.62 |
| **Geneformer+LGBM** | 63.7 |
| **Cell2Text-Llama-1B-LoRa** | 75.57 |
| **Cell2Text-Llama-1B** | 85.31 |
| **Cell2Text-Gemma-4B** | **85.62** |

### 4.2.4 PATHWAY ACTIVITY IDENTIFICATION

Beyond cellular identity, we assessed the models' ability to identify active biological processes by classifying pathway enrichments. This task is framed as a multi-label classification problem where the goal is to identify the top two active pathways from a predefined set of 34 Hallmark pathways. A detailed description of the evaluation metrics is provided in Appendix E.2.

Table 4 demonstrates a notable trade-off between the classifiers and our generative Cell2Text framework. Cell2Text models show competitive performance on these measures while surpassing the Geneformer+Head baseline. This represents an important observation: although trained primarily to generate coherent textual descriptions, Cell2Text exhibits strong classification performance as a secondary capability. The Geneformer+LGBM performs relatively better on the ranking-based evaluation metrics due to more sophisticated candidate-wise binary classification setup. Our model delivers good predictive results without explicit optimization for this particular task, demonstrating the rich representational capacity of its learned features.

Table 4: Pathway classification performance showing Cell2Text models achieve good results across diverse metrics despite not being specialized for this task. Subset Accuracy (Acc), Jaccard similarity (Jac), Weighted F1 (F1)

|  | Acc | Jac | F1 |
| --- | --- | --- | --- |
| **Geneformer+Head** | 40.08 | 57.22 | 63.60 |
| **Geneformer+LGBM** | **44.09** | **60.39** | **67.11** |
| **Cell2Text-Llama-1B-LoRa** | 39.63 | 56.73 | 64.06 |
| **Cell2Text-Llama-1B** | 42.31 | 58.76 | 66.16 |
| **Cell2Text-Gemma-4B** | 42.19 | 58.67 | 66.27 |

## 5 CONCLUSION

In this work, we presented Cell2Text, a multimodal generative framework that generates interpretable natural language descriptions from single-cell expression data. By combining gene-level embeddings from pretrained single-cell foundation models with instruction-tuned language models, our approach generates biologically meaningful cell text descriptions and achieves competitive classification performance for cellular identity, tissue context, and pathway activity. Our model outperforms specialized baselines on cell type, tissue, and disease prediction tasks, while maintaining high semantic fidelity, suggesting that training for text generation creates richer cellular representations than traditional classification approaches. Our PageRank-based evaluation further reveals that the model's prediction is ontologically coherent with minimal error. More broadly, the integration of biological domain–specific pretrained models with large language models offers a general strategy for building scalable and interpretable frameworks that can extend beyond cell annotation to broader challenges in computational biology.

**Reproducibility Statement:** For reproducibility, hyperparameters are detailed in Appendix D and our anonymized codebase is available at `https://anonymous.4open.science/r/cell2text-FDDF`.

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

# A  DATASET CONSTRUCTION DETAILS

## A.1  SAMPLING METHODOLOGY

Our sampling strategy addresses three key challenges in large-scale single-cell dataset construction: extreme class imbalance across cell types and tissues, batch effects introduced by different studies and sequencing technologies, and the need for statistically rigorous train–validation–test splits that prevent data leakage.

To construct a dataset suitable for robust model training, we implemented a principled sampling framework designed to mitigate biases inherent in aggregated public data and enhance biological diversity. Public datasets are often dominated by a few common tissues (e.g., blood, brain), cell types, and disease conditions, which can skew model learning. Our approach addresses this by creating a balanced cohort through a composite, multi-objective stratification strategy.

| Variable | Before Sampling | After Sampling |
|---|---|---|
| Cell Type | 0.7470 | 0.8431 |
| Tissue (general) | 0.6106 | 0.7035 |
| Disease | 0.3479 | 0.4957 |

Table 5: Normalized Shannon diversity before and after applying the sampling strategy.

To reduce assay-specific confounding, we excluded a subset of protocols that were either rare, highly heterogeneous, or not directly comparable to standard droplet-based transcriptome profiling. Specifically, we removed full-length assays such as the Smart-seq family, which differ substantially in coverage and sensitivity; niche or proprietary protocols (e.g., Quartz-seq, GEXSCOPE) with limited adoption; and targeted assays such as BD Rhapsody targeted mRNA, which do not capture the full transcriptome. We also excluded very low-prevalence technologies, including 10x Flex, to avoid unstable representation. By focusing on well-represented droplet-based and complementary protocols, the resulting dataset maintains diversity across major assay families while minimizing technical biases that could obscure biological signal.

We implement donor-level splitting with an 80/10/10 ratio, guaranteeing that no individual donor contributes to multiple splits while maintaining broad representation of biological categories across all partitions.

## A.2  PATHWAY ACTIVITY ANALYSIS

We compute pathway activity scores using pySCENIC (Aibar et al., 2017), evaluating 50 curated pathway signatures from the MSigDB Hallmark collection (Liberzon et al., 2011). Prior to pathway scoring, we perform global highly variable gene (HVG) selection using the Seurat method (Satija et al., 2015) across the dataset to reduce noise and dimensionality, ensuring that enrichment is computed on informative genes while preserving biological variability.

pySCENIC then calculates enrichment scores for each cell–pathway pair using the AUCell algorithm, which ranks genes within each cell by expression level and computes the area under the curve (AUC) for genes in each pathway signature, providing a quantitative measure of pathway activity.

Pathways active in fewer than 0.5% of cells are filtered to retain only biologically meaningful processes, resulting in 34 pathways. This threshold ensures that retained pathways represent genuine biological signals rather than noise while maintaining sufficient diversity of functional annotations. For each cell, we identify the two most enriched pathways to capture the primary biological processes while ensuring computational efficiency.

## A.3 TEXT DESCRIPTION GENERATION

Natural language descriptions are constructed by integrating multiple information sources: cell type metadata from CELLxGENE Census (Program et al., 2024), standardized ontology annotations from the Cell Ontology (OBO Foundry) (Smith et al., 2007), and functional context from pathway activity analysis.

Cell type information is standardized using Cell Ontology terms, which provide consistent definitions, synonyms, and hierarchical relationships. This standardization ensures consistent terminology across different studies and enables semantic understanding of cellular identities.

An example description of a cell type generated using this approach is provided in Example A.3.

> *This sample consists of a ciliated columnar cell of tracheobronchial tree, multi-ciliated epithelial cell located in the trachea and bronchi, characterized by a columnar shape and motile cilia on its apical surface. These cilia facilitate mucociliary clearance by moving mucus and trapped particles toward the pharynx. It originates from the lung parenchyma of a normal male during elderly stage. This cell is associated with Genes mediating programmed cell death (apoptosis) by activation of caspases. Additionally, it involves Genes down-regulated in response to ultraviolet (UV) radiation.*

## B DATASET STATISTICS AND DISTRIBUTIONS

Figure 2 depicts the token length distribution of gene expression sequences post-tokenization with the Geneformer tokenizer, averaging 1843.2 tokens. The distribution peaks between 1000-1500 tokens, tapering off, with a spike at the 4096-token maximum, indicating some sequences are adjusted to this limit.

Figure 3 illustrates the token length distribution of natural language descriptions tokenized with the Llama-3.2-1B-Instruct tokenizer, with a mean length of 104.3 tokens. The distribution peaks around 100-150 tokens and decreases steadily, with fewer descriptions exceeding 200 tokens.

Figure 4 shows a skewed distribution of cell types, with glial cells like oligodendrocytes dominating due to abundant brain-derived data, while immune cells like T and B cells are also prominent, reflecting bias toward easily accessible lymphoid tissues. Rarer types, such as stromal and plasma cells, are underrepresented, likely due to challenges in cell isolation and lower natural prevalence, highlighting how dataset composition reflects methodological biases rather than just biology.

Figure 5 shows a skewed distribution of the top 30 disease categories (out of 128), with normal (592,932) and COVID-19 (61,961) dominating. The high count for normal likely stems from extensive use of healthy control samples in research to establish baselines, while COVID-19's prominence reflects widespread data collection during the pandemic.

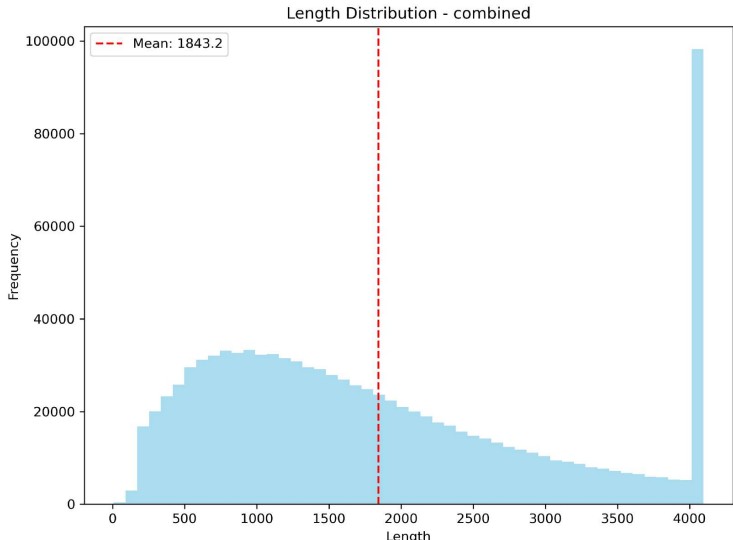

Figure 2: Token length distribution of gene expression sequences after tokenization with the Geneformer tokenizer.

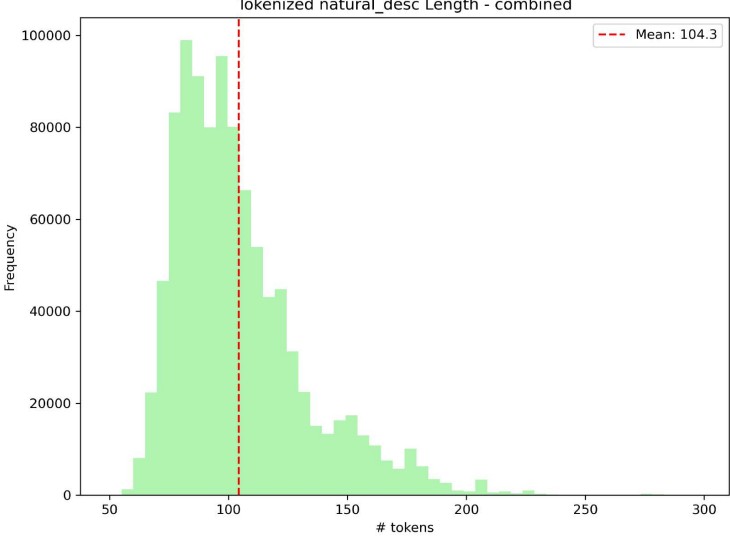

Figure 3: Token length distribution of natural language descriptions after tokenization with the Llama-3.2-1B-Instruct tokenizer.

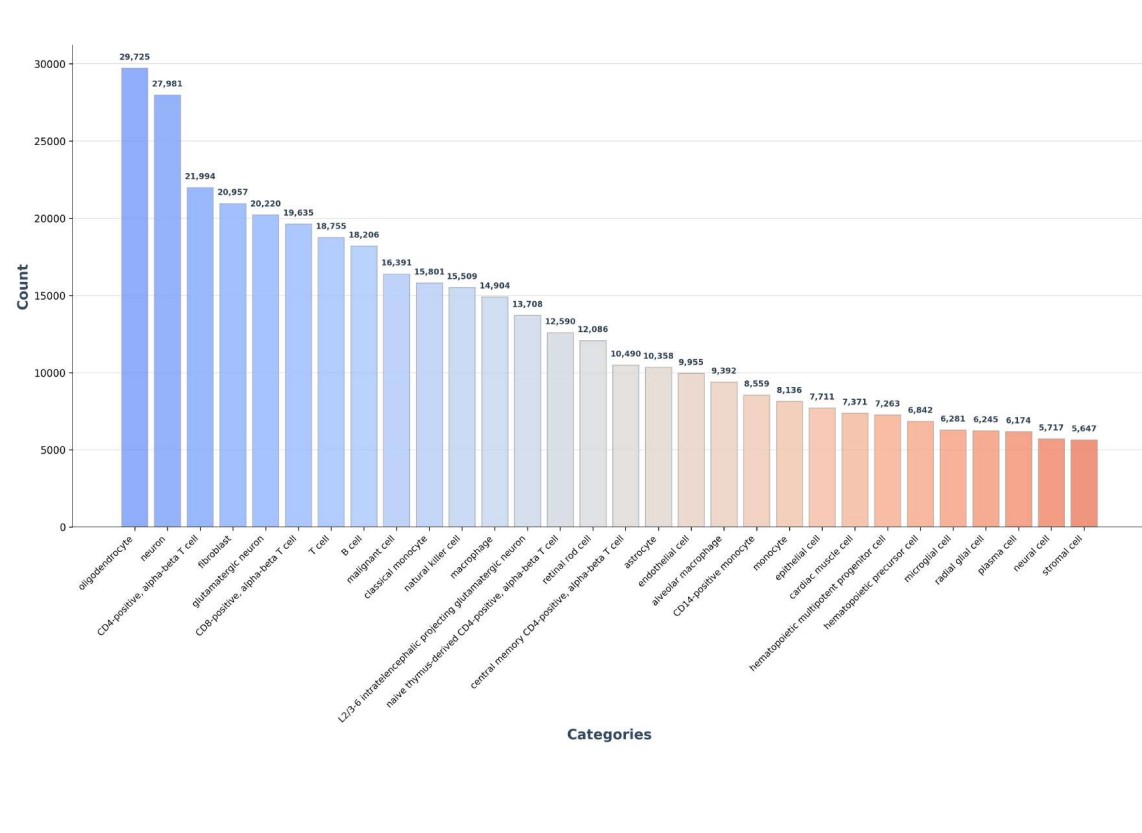

Figure 4: Overview of the distribution of cell types in the dataset. For clarity, only the 30 most abundant categories out of 783 are shown.

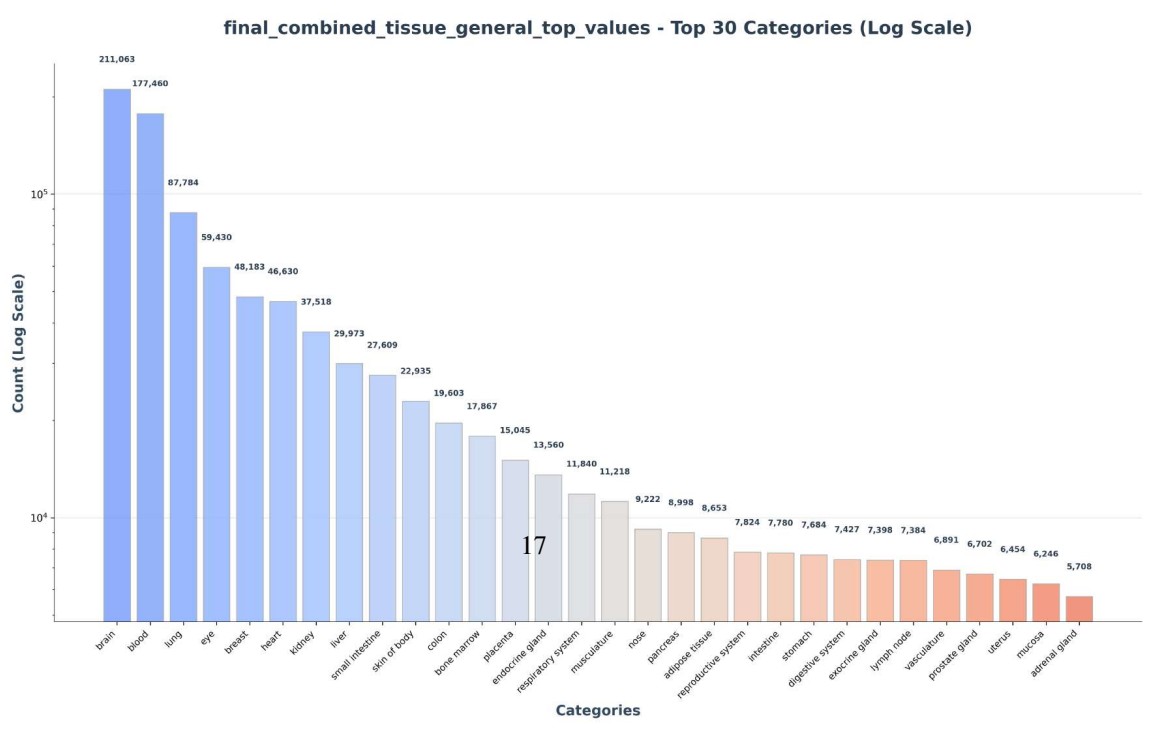

Figure 6: Overview of the distribution of tissue (general) categories in the dataset. For clarity, only the 30

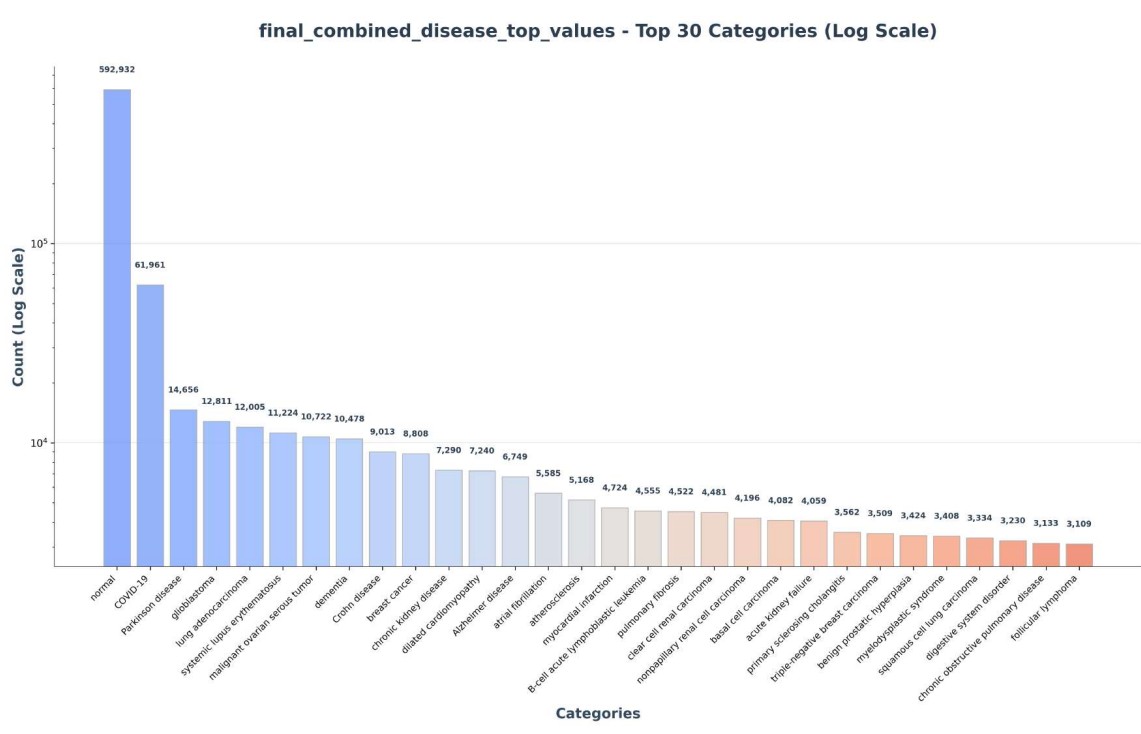

Figure 5: Overview of the distribution of disease categories in the dataset. For clarity, only the 30 most abundant categories out of 128 are shown.

Figure 6 shows a skewed distribution of the top 30 tissue categories (out of 347), with brain (211,063) and blood (177,460) leading, likely due to extensive sampling in neurological and hematological research. Tissues like lung (87,784) and eye (59,430) follow, reflecting biases toward accessible or clinically relevant sources.

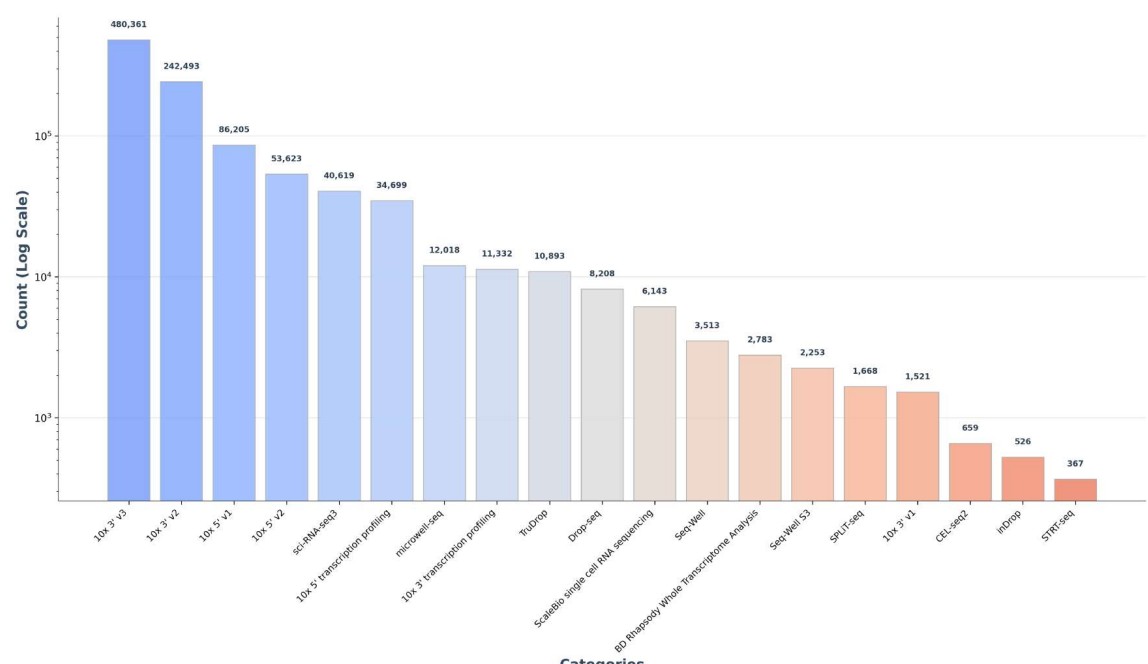

Figure 7: Overview of the distribution of assay categories in the dataset.

Figure 7 shows a skewed distribution of assay categories, with 10x v3 (480,361) and 10x v2 (242,493) dominating, likely due to their widespread adoption in droplet-based single-cell RNA sequencing. Assays like scRNA-seq3 (86,205) and 10x transcription profiling (53,623) follow, reflecting a bias toward scalable, standardized protocols.

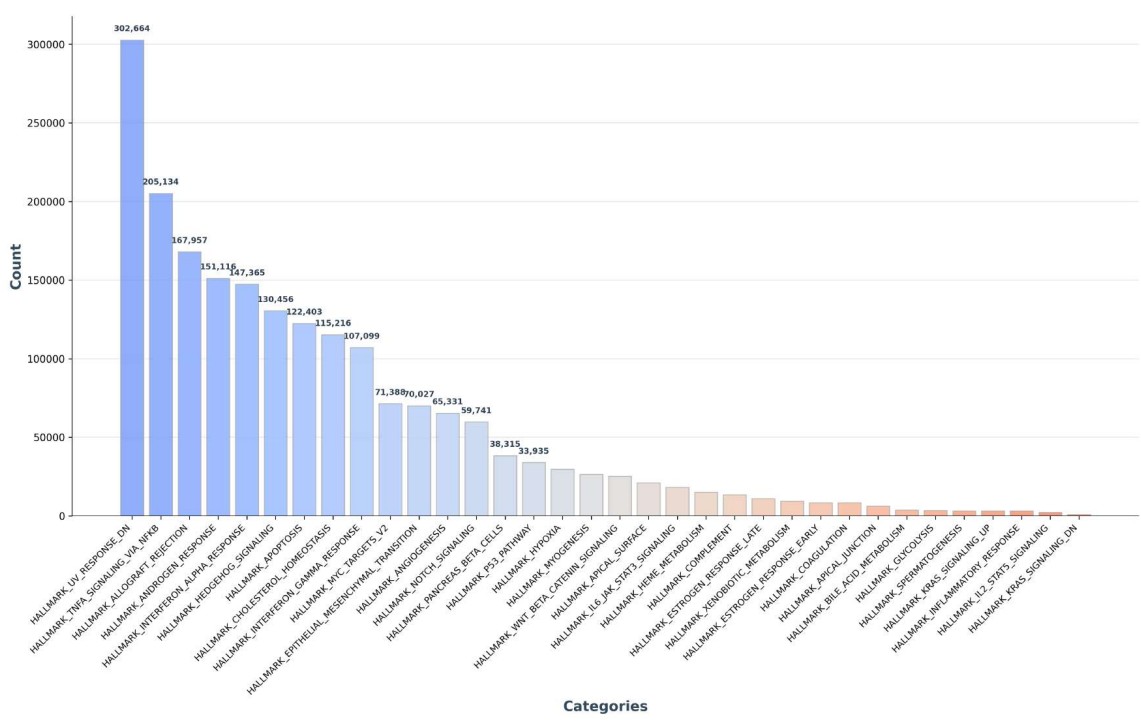

Figure 8: Overview of the distribution of pathway categories in the dataset.

Figure 8 shows a skewed distribution, with hallmark TNF-$\alpha$ via NF$\kappa$B (302,664) and hallmark UV response DN (205,134) leading, likely due to inflammation and stress studies, boosted by brain and blood tissue dominance (Figure 6). Pathways like hallmark interferon-$\alpha$ response (167,957) and hallmark allograft rejection (151,116) follow, reflecting immune biases from lymphoid samples. Rare pathways like hallmark spermatogenesis (3,109) are underrepresented, possibly due to tissue specificity.

## C  CELL TYPE SIMILARITY DISTRIBUTION

The similarity scores, computed using Personalized PageRank on the Cell Ontology graph, exhibit a characteristic distribution that validates their utility. As shown in Figure 9, the distribution of similarity scores is highly skewed, with the vast majority of cell type pairs having a similarity value close to zero, reflecting the sparse and hierarchical nature of the ontology where most cell types are distantly related.

Quantitatively, the similarity scores range from 0 to 1, with a mean of 0.049 and median of 0.016, confirming the heavy-tailed nature of the distribution. The low median relative to the mean (0.016 vs 0.049) indicates strong right skewness. Notably, 95% of cell type pairs have similarity scores below 0.215, while only the top 1% of pairs achieve similarities above 0.438, demonstrating that truly related cell types are rare and easily distinguished from the majority of unrelated pairs.

The cumulative distribution function (CDF) in Figure 10 further illustrates this property, with the sharp rise in the curve at low similarity values confirming that a large fraction of pairwise similarities are small. This heavy-tailed nature of the distribution is crucial, as it demonstrates the metric's ability to effectively discriminate between the few closely related cell types and the many unrelated ones, which is essential for our nuanced evaluation of model predictions.

Statistical analysis confirms the heavy-tailed nature of our similarity distribution. Log-log regression analysis shows strong linearity ($R^2 = 0.862$) with a power-law exponent of $\alpha = 0.67$, while rank-frequency analysis demonstrates excellent fit quality ($R^2 = 0.930$). These results validate that our PageRank-based similarities exhibit the expected heavy-tailed characteristics of hierarchical biological networks, ensuring effective discrimination between closely related and distant cell types.

| Statistic | Value |
|---|---|
| Mean | 0.049 |
| Median | 0.016 |
| Standard Deviation | 0.087 |
| 95th Percentile | 0.215 |
| 99th Percentile | 0.438 |

Table 6: Summary statistics for Cell Ontology PageRank similarity scores across all cell type pairs.

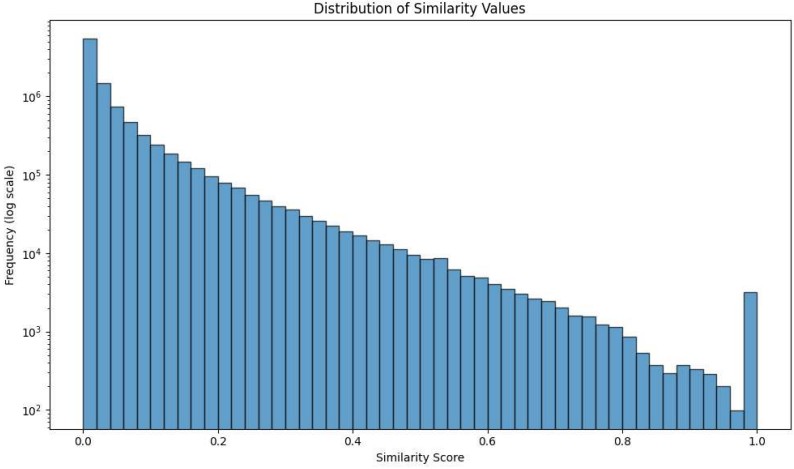

Figure 9: Distribution of similarity scores across all cell type pairs, with a logarithmic frequency scale.

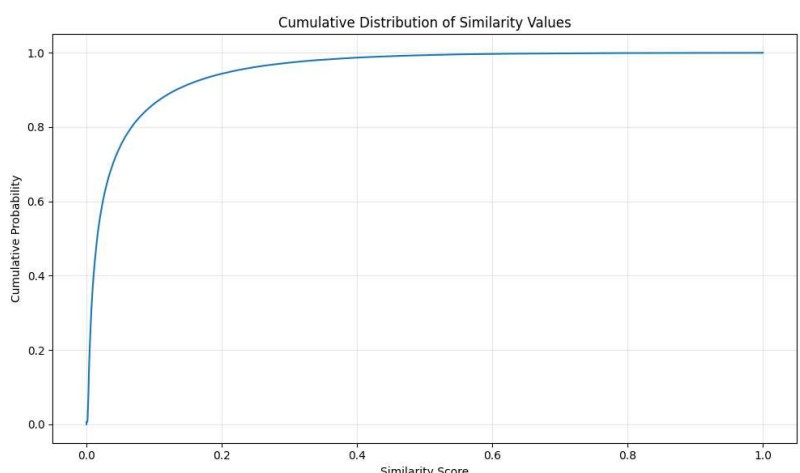

Figure 10: Cumulative distribution of similarity scores.

## D  HYPERPARAMETERS AND TRAINING DETAILS

### D.0.1  CELL2TEXT HYPERPARAMETERS

**Cell2Text-Llama-1B-LoRA**: Our model is implemented using PyTorch and trained on a single node with 8 NVIDIA V100 32GB GPUs. We use the Adam (Adam et al., 2014) optimizer with a learning rate of 0.0002 and StepLR scheduler with $\gamma = 0.98$ that decays the learning rate every epoch. For the LoRA adapter, we apply it exclusively to the self-attention modules in the LLaMA decoder, using a rank of 256 and an $\alpha$ value of 512. Training lasts for 3 epochs of supervised fine-tuning. The batch size is set to 2 per GPU, and gradient accumulation is applied every 8 forward passes, resulting in an effective batch size of 128.

**Cell2Text-Llama-1B**: The full fine-tuning variant is implemented using PyTorch and trained on a single node with 8 NVIDIA V100 32GB GPUs. We use the Adam optimizer with a learning rate of 0.0002 and StepLR scheduler with $\gamma = 0.98$ that decays the learning rate every epoch. Training lasts for 2 epochs of supervised fine-tuning. The batch size is set to 3 per GPU, and gradient accumulation is applied every 8 forward passes, resulting in an effective batch size of 192.

**Cell2Text-Gemma-4B**: Our model is implemented using PyTorch and trained on a single node with 8 NVIDIA A100 80GB GPUs. We use the Adam optimizer with a learning rate of 0.00005 and StepLR scheduler with $\gamma = 0.98$ that decays the learning rate every epoch. Training lasts for 3 epochs of supervised fine-tuning. The batch size is set to 2 per GPU, and gradient accumulation is applied every 8 forward passes, resulting in an effective batch size of 128.

### D.0.2  BASELINES HYPERPARAMETERS

**Geneformer+Head:** We trained the model using the AdamW optimizer with a weight decay of $0.01$ and an initial learning rate of $5 \times 10^{-5}$. Training was conducted for 3 epochs with a batch size of $64$ per GPU, employing gradient clipping at a maximum norm of $1.0$ and automatic mixed precision (AMP) to

enhance stability and efficiency. All experiments were run with distributed data parallelism (DDP) across two NVIDIA A6000 GPUs (48 GB each).

**LGBM for pathway classification :** we trained the LightGBM with a binary objective ($objective =$ "$binary$") optimized using log loss as the evaluation metric. Each classifier was trained with a maximum of 1000 boosting iterations ($n\_estimators = 1000$), with early stopping (patience = 50) to prevent overfitting. We used a learning rate of 0.05, balancing training stability with convergence speed, and limited the model complexity by setting the maximum number of leaf nodes to 31 ($num\_leaves = 31$).

**LGBM for other classification task :** we trained a LightGBM classifier with a multiclass objective (objective="multiclass") and multi-class log loss ($metric =$ "$multi\_logloss$") as the evaluation metric. We used a maximum of 2000 boosting iterations ($n\_estimators = 2000$) with early stopping ($patience = 100$) to avoid overfitting, guided by performance on the validation set. A learning rate of 0.05 was chosen to balance convergence speed with generalization, and the tree complexity was controlled by setting the maximum number of leaf nodes to 31 ($num\_leaves = 31$).

# E  ADDITIONAL EVALUATION DETAILS

## E.1  EVALUATION METRICS FOR TEXT GENERATION

For text generation, we employ metrics that capture both surface-level similarity and semantic fidelity between generated and reference texts:

- **Exact Match (Exct)**: A strict lower bound that assigns a score of 1 only if the generated text exactly matches the reference string character-for-character; otherwise 0. Averaged over the dataset.

$$\text{Exct} = \frac{1}{N} \sum_{i=1}^{N} \mathbf{1}\{y_i^{\text{gen}} = y_i^{\text{ref}}\}$$

- **BLEU (B-2, B-4)**: Measures $n$-gram precision with a brevity penalty, rewarding overlap between generated and reference tokens. For BLEU-$n$, precision is computed over all $n$-grams:

$$\text{BLEU-}n = \text{BP} \cdot \exp\left(\sum_{k=1}^{n} w_k \log p_k\right),$$

  where $p_k$ is the modified $k$-gram precision, $w_k$ are uniform weights, and BP is the brevity penalty.

- **ROUGE (R-1, R-2, R-L)**: Measures recall of overlapping units (unigrams, bigrams, or longest common subsequence) between generated and reference text:

$$\text{ROUGE-n} = \frac{\sum_{\text{ngram} \in y^{\text{ref}}} \min\big(\text{Count}_{y^{\text{gen}}}(\text{ngram}), \text{Count}_{y^{\text{ref}}}(\text{ngram})\big)}{\sum_{\text{ngram} \in y^{\text{ref}}} \text{Count}_{y^{\text{ref}}}(\text{ngram})}.$$

- **BERTScore** (Zhang et al., 2020): Computes semantic similarity by aligning each token embedding in the generated text to its most similar token embedding in the reference text using contextual embeddings. We report F1-scores:

$$\text{BERTScore-F1} = 2 \cdot \frac{\text{Precision} \cdot \text{Recall}}{\text{Precision} + \text{Recall}}.$$

  We use two pretrained encoders: **RoBERTa (RBT-f1)** for general language understanding and **BioBERT (BBT-f1)** (Lee et al., 2020), which specializes in biomedical semantics.

### E.2 EVALUATION METRICS FOR PATHWAY ACTIVITY IDENTIFICATION

For pathway activity classification, we employ a comprehensive suite of metrics:

- **Accuracy (Subset Accuracy)**: The strictest metric, which counts a prediction as correct only if the predicted set of pathways exactly matches the true set. Formally:

$$\text{Acc} = \frac{1}{N} \sum_{i=1}^{N} \mathbf{1}\{\hat{Y}_i = Y_i\},$$

  where $Y_i$ is the true set of pathways for sample $i$, and $\hat{Y}_i$ is the predicted set.

- **Jaccard Similarity**: A softer metric that measures the intersection-over-union (IoU) of predicted and true pathway sets:

$$\text{Jac} = \frac{1}{N} \sum_{i=1}^{N} \frac{|\hat{Y}_i \cap Y_i|}{|\hat{Y}_i \cup Y_i|}.$$

- **F1-Score (Weighted)**: The weighted average of per-class F1-scores, where weights are proportional to class frequency:

$$\text{F1}_{\text{weighted}} = \sum_{c \in \mathcal{C}} \frac{|Y_c|}{\sum_{c' \in \mathcal{C}} |Y_{c'}|} \cdot \text{F1}_c,$$

  with $\text{F1}_c = \frac{2 \cdot \text{Prec}_c \cdot \text{Rec}_c}{\text{Prec}_c + \text{Rec}_c}$.

## F  LLM USAGE

Large language models were used for reformulation and refinement of the paper text to improve clarity and readability. It was not used for research ideation, methodological design, data analysis or the discovery of scientific insights.

