# OpenReview forum: "Cell2Text: Multimodal LLM for Generating Single-Cell Descriptions from RNA-Seq Data"
_ICLR.cc/2026/Conference — Submitted to ICLR 2026_

### Official Review · Reviewer_a2MT · 2025-10-29

**Soundness:** 2
**Presentation:** 2
**Contribution:** 1
**Rating:** 2
**Confidence:** 4

**Summary:**

The paper proposes Cell2Text, a multimodal generative system that converts single-cell RNA-seq profiles into structured natural language summaries, addressing the limits of discrete prediction heads that reduce cellular complexity to fixed labels. The method fuses gene-level embeddings from single-cell foundation models with a pretrained language model to describe cell identity, tissue context, disease links, and pathway activity, and it generalizes to unseen cells. Experiments show higher classification accuracy than baselines, strong alignment with ontologies measured by a PageRank-style similarity, and high semantic fidelity in the generated text.

**Strengths:**

1. The related work is broad and up to date.
2. The idea of turning gene embeddings into textual summaries is interesting.

**Weaknesses:**

1. The paper does not establish why textual summaries are needed for biological tasks. A lot models can perform well on cell annotation tasks, and I don’t think generating text from a model of billions of parameters then do the classification is a better way compared to lightweight annotation models like CellTypist. For pathway activity identification task, the proposed model underperforms Geneformer while using a larger backbone, which further confuses the claim.
2. Each of the cell annotation and pathway activity identification tasks is evaluated on a single dataset. It is insufficient to draw any useful conclusion from that.
3. For cell annotation tasks, please provide the results of other simple and established baselines. In addition, it would be useful to see the comparison of training and inference cost between the proposed method and the other baselines.

**Questions:**

1. In what concrete research workflows would textual summaries improve biological discovery?
2. How sensitive are results to the choice of the underlying language model and to the choice of the gene embedding model?
3. Human has 50 Hallmark pathways. Why only 34 of them are selected to perform pathway activity identification?

---

### Official Review · Reviewer_GCbo · 2025-10-31

**Soundness:** 2
**Presentation:** 2
**Contribution:** 2
**Rating:** 4
**Confidence:** 3

**Summary:**

This paper proposes Cell2Text, a multimodal framework that aligns per-gene embeddings from a frozen single‑cell encoder (Geneformer) with an instruction‑tuned LLM via a lightweight adapter to generate structured natural‑language cell descriptions capturing type, tissue, disease, and pathway activity, and evaluates both text quality and label extraction against Geneformer‑based classifier baselines on a 1M‑cell CELLxGENE‑derived corpus. The method preserves per‑gene embeddings rather than pooled cell vectors, projects them into the LLM token space with a two‑layer adapter (with L2 normalization), and decodes descriptions using Llama‑3.2‑1B or Gemma‑3‑4B under instruction prompts, while the encoder (Geneformer) stays frozen.

**Strengths:**

- Clear, modular architecture that leverages strong domain encoders and compact instruction‑tuned LLMs, with rationale for per‑gene tokens and freezing the cell encoder to preserve pretrained structure.

- Well‑motivated ontology‑aware evaluation using Personalized PageRank on the Cell Ontology to quantify near‑miss biological closeness rather than flat accuracy alone.

- The code was made available during the review process, which shows the transparency of the authors

**Weaknesses:**

- Limited ablations isolating which components drive gains (e.g., per‑gene tokens vs pooling, adapter depth, normalization, freezing choices) and how much the LLM contributes beyond a simpler decoder head conditioned on the same embeddings.

- Reliance on generated descriptions whose supervision partially comes from metadata and pySCENIC pathway calls may encode dataset biases; it is unclear how robust the model is to misannotations, rare types, and cross‑assay shifts beyond the reported sampling controls.[1]

- Evaluation extracts discrete labels via regex from free text; noise in extraction could confound the reported accuracy gains vs. training a discriminative head on the same supervision, and error analysis by category frequency is limited in the main text.[1]

**Questions:**

- Why freeze the Geneformer encoder entirely rather than exploring partial unfreezing (e.g., top layers) or adapter‑based updates (eg, LoRA) in the encoder to better align with the LLM while preserving pretraining? Was this compared empirically?

- What drove the two‑layer adapter choice and its dimensionality? were deeper adapters, residual adapters, or attention pooling over gene tokens into a small set of learned “prefix tokens” benchmarked for stability and compute ?

- The sequence length caps at 4,096 genes; how are genes selected and ordered for cells expressing more than this cap, and what is the sensitivity to gene ordering (e.g., by expression rank vs canonical tokenizer order)?

- Why choose Llama‑1B and Gemma‑4B? Were stronger small LLMs or domain LMs (e.g., finetuned for the biomedical domain) considered for evaluation? And what is the scaling curve vs parameter count under the same compute budget?

- How do errors or inconsistencies in metadata and pathway inference propagate to the generative model, and is there a robustness analysis under corrupted labels or across assays left out of training?

- Are pathway outputs evaluated beyond the top‑2 constraint used to generate text? Can the model recover multi‑pathway activity profiles if prompted? And how does performance change when supervising with more than two pathways per cell?

- For extracted labels, why compare primarily against a linear head and LightGBM on pooled Geneformer embeddings? Can a token‑wise attentive classifier over per‑gene embeddings or a shallow transformer head close the performance gap between the simple classifiers and the LLM-decoders?

- It’s hard to appreciate the contributions of this work on the classification tasks without comparison to other tools. Have you considered benchmarking against recent single‑cell foundation models that expose gene‑token interfaces (e.g., scGPT, scBERT with token outputs) using similar training data and frozen settings to isolate the effect of the decoder?

- The PageRank similarity is compelling, but can the authors report calibration of similarity vs human expert judgments and show confusion matrices highlighting near‑misses across ontology depth levels?

- Regarding the BERTScore metrics shown, do human evaluations confirm factual correctness of biological claims in generated descriptions (e.g., pathways consistent with known markers) and measure hallucination rates?

---

### Official Review · Reviewer_EsKz · 2025-10-31

**Soundness:** 3
**Presentation:** 3
**Contribution:** 2
**Rating:** 6
**Confidence:** 4

**Summary:**

This paper presents Cell2Text, a framework that integrates single-cell gene expression data with large language models (LLMs). The method uses gene-level embeddings to represent cells and conditions an LLM to process or generate text from these embeddings. The goal is to capture more nuanced biological information compared to methods relying on cell-wide embeddings, or compress gene-level representation.

**Strengths:**

•	Interesting use of Personalized PageRank (PPR) to achieve a more nuanced evaluation of cell-type classification, addressing how LLMs can struggle with categorical outputs.
•	The idea of gene-based embeddings instead of full cell-wide representations is well motivated and potentially captures finer biological detail.

**Weaknesses:**

•	The paper does not compare against Cell2Sentence, which already includes text-based cell representations.
•	The paper makes the point of the benefit of the gene-embedding representation but does not empirically that is actually better than using the top X genes, or text-based representation. While the idea of gene-level embeddings is interesting, the paper does not discuss whether this design leads to measurable gains over simpler input choices

**Questions:**

1.	Since PPR is only used for evaluation, have you considered incorporating a similar objective during training to allow flexibility for biologically similar errors?
2. Why wasn’t Cell2Sentence included as a baseline for comparison?

---

### Official Review · Reviewer_k6Tb · 2025-11-03

**Soundness:** 1
**Presentation:** 2
**Contribution:** 2
**Rating:** 2
**Confidence:** 5

**Summary:**

This paper introduces Cell2Text, a multimodal generative framework that produces interpretable natural language descriptions from single-cell RNA expression data. By integrating gene-level embeddings from geneformer with instruction-tuned language models.

**Strengths:**

1. Exploring LLMs in single-cell omics data is promissing direction.

2. The annotated 1M cells enriched with ontology terms, tissue and disease metadata, and pathway annotations may be useful to the community.

**Weaknesses:**

1. The current design, which feeds gene embeddings directly into the system prompt of the LLM, lacks a clear biological or computational rationale. Since the system prompt is meant for textual instructions rather than numerical embeddings, this configuration is not meaningful. Moreover, the “Sequence embeddings” input format does not specify which embeddings correspond to which genes, making it impossible for the LLM to infer the biological context or interpret the underlying signals. A more explicit mapping or token-level alignment between gene identifiers and embeddings would be required for interpretability and reproducibility.

2. Restricting the base language model to Meta-Llama-3.2-1B-Instruct and Gemma3-4B-it significantly limits the expressive capacity and reasoning ability of the framework. Larger models, such as Llama-3.1-70B, Gemma-27B, or equivalent-scale open-weight LLMs—could substantially enhance biological text generation and cross-modal understanding. The architecture should therefore be designed to support flexible integration with larger instruction-tuned models to fully exploit the potential of multimodal learning.

3. Text Generation Evaluation Scope
The evaluation of text generation quality using broad metrics such as BLEU, ROUGE, and BERTScore is overly general. Instead, it would be more informative to benchmark specific task-oriented generation capabilities, such as functional annotation summarization, disease association reasoning, or pathway interpretation. Furthermore, cell description generation need not be constrained to a single textual annotation per cell—multiple complementary textual outputs (e.g., morphology-focused, pathway-focused, or disease-context summaries) could capture richer biological semantics.

4. Baselines for Cell Type Annotation
The comparison with baseline models omits key categories of state-of-the-art methods:

Traditional annotation models: such as ACTINN, CellTypist, and scDeepSort, which remain strong supervised benchmarks for cell type identification.

LLM-based approaches: including LangCell, CellWhisperer, CELLama, Cell2Sentence, and Cell2Sentence-Scale, which represent the most relevant recent multimodal and text-generative frameworks in this domain.
Including these baselines would provide a more comprehensive and credible performance comparison.

**Questions:**

1. Leveraging LLMs for single-cell transcriptomic data should take advantage of their large context windows to encode all expressed genes, rather than limiting input to 4,096 genes. Highly expressed genes are not necessarily biologically meaningful, like housekeeping genes, for instance, can dominate expression without contributing to cellular identity.

2. The claim that a two-layer feedforward network with non-linear activation can preserve biological nuances lacks theoretical justification. It is unclear how such a simple projection adequately maps complex gene-level embeddings into the natural language semantic space.

3. The output of the modality adapter is ambiguous. If the input consists of 4,096 gene embeddings, does the adapter output the same number of projected embeddings, or a single aggregated representation? This requires clarification.

4. The choice of relatively small models—Meta-Llama-3.2-1B-Instruct and Gemma3-4B-it—is restrictive. Larger instruction-tuned models like Llama-3.1-8B or Llama-3.1-70B would likely provide greater reasoning capacity and biological interpretability.

5. The Geneformer and the LLM do not share a common tokenizer or embedding space, leading to significant distributional mismatch. The proposed modality adapter alone may not be sufficient to fully bridge this gap.

6. It is conceptually flawed to input embeddings into the LLM’s system prompt. Moreover, using “Sequence embeddings:” without explicit gene identifiers prevents the LLM from understanding which genes the embeddings represent.

---

### Meta-Review · Area_Chair_49pX · 2025-12-19

**Summary:**

Reviewers questioned whether injecting gene-level embeddings into an instruction-tuned LLM meaningfully leverages either biological structure or language model capabilities in a principled way. While the idea of producing textual cell descriptions is interesting, the current formulation was widely viewed as insufficiently grounded and only weakly differentiated from simpler or existing approaches. Overall, the contribution was judged to fall short of the rigor and clarity expected for acceptance.

**Reviewer Concerns:**

Because no rebuttal was provided, none of the major concerns were addressed.

**Reviewer Scores:**

Because no rebuttal was provided, the score distribution would likely tighten around rejection.

---

### Decision · Program_Chairs · 2026-01-26

Reject